# Future climate change shaped by inter-model differences in Atlantic meridional overturning circulation response

Katinka Bellomo [1]✉, Michela Angeloni[2,1], Susanna Corti[3] & Jost von Hardenberg[4,1]

In climate model simulations of future climate change, the Atlantic Meridional Overturning Circulation (AMOC) is projected to decline. However, the impacts of this decline, relative to other changes, remain to be identified. Here we address this problem by analyzing 30 idealized abrupt-4xCO$_2$ climate model simulations. We find that in models with larger AMOC decline, there is a minimum warming in the North Atlantic, a southward displacement of the Inter-tropical Convergence Zone, and a poleward shift of the mid-latitude jet. The changes in the models with smaller AMOC decline are drastically different: there is a relatively larger warming in the North Atlantic, the precipitation response exhibits a wet-get-wetter, dry-get-drier pattern, and there are smaller displacements of the mid-latitude jet. Our study indicates that the AMOC is a major source of inter-model uncertainty, and continued observational efforts are needed to constrain the AMOC response in future climate change.

[1] National Research Council of Italy, Institute of Atmospheric Sciences and Climate (CNR-ISAC), Turin, Italy. [2] Department of Physics and Astronomy, Alma Mater Studiorum - University of Bologna, Bologna, Italy. [3] National Research Council of Italy, Institute of Atmospheric Sciences and Climate (CNR-ISAC), Bologna, Italy. [4] Department of Environment, Land and Infrastructure Engineering, Politecnico di Torino, Turin, Italy. ✉email: k.bellomo@isac.cnr.it

The Atlantic meridional overturning circulation (AMOC) is a large system of ocean currents, and plays a crucial role in the earth's climate by regulating the global transport of heat, carbon, and freshwater. It is estimated that annually as much as ~0.5 PW of heat is carried across the equator into the North Atlantic by the AMOC, which is believed to be responsible for making the northern hemisphere ~1 °C warmer than the southern hemisphere[1–3] and for shifting the zonal average of the Inter-tropical convergence zone (ITCZ) north of the equator at about 5°N, thus influencing global rainfall and atmospheric circulation patterns[4,5].

Not only the AMOC is central to the earth's climate, but it has also been identified as one of the tipping elements in the earth system[6]. In fact, abrupt changes in the AMOC have been implicated in glacial-interglacial transitions[7–9], such as Dansgaard–Oeschger oscillations[10]. The role of the AMOC in amplifying these transitions is supported by deep-ocean proxy data[11,12], while the potential impacts of abrupt AMOC shut-downs have been examined in idealized climate model simulations in which the AMOC is artificially halted. These simulations, often referred to as water hosing experiments, show that an AMOC shutdown causes cooling of the northern hemisphere by several degrees, increased sea ice in the North Atlantic and Nordic Sea, and a southward shift of the ITCZ[13–15]. Even smaller AMOC declines have been shown to cause widespread impacts[16,17].

Since there is increasing evidence that the AMOC may have slowed down over the last century[18,19], understanding how the AMOC may change in the future is of primary interest, including the possibility that the AMOC may reach a tipping point threshold[3]. Direct observations to monitor the AMOC, which began in 2004 with the RAPID-MOCHA array[3,20], also show a decline[21–23], although internal variability is large and the observed time period is too short to estimate a trend[24–26]. From these observations it is not yet possible to quantify the anthropogenic contribution to the AMOC decline; however, model projections of future climate change show a further decline of the AMOC into the 21st century in response to greenhouse gas forcing[27,28]. This decline has been shown to be related to a decrease in the density of sea water in the subpolar North Atlantic (SPNA)[29–31], and is associated with a reduced warming in the SPNA sea surface temperature (SST), often referred to as the North Atlantic warming hole (NAWH)[17,19].

Through the NAWH, the AMOC effectively counteracts the simulated 21st century warming over the North Atlantic basin[32–35]. A recent study[36] has investigated the role of atmospheric processes that could alone lead to the formation of the NAWH. Land–sea warming contrast may also explain in part the temperature anomaly difference[37]. Nevertheless, it seems that in model projections of future climate change, ocean circulation plays the largest role leading to the onset and development of the NAWH[17,36,38,39]. The NAWH, by changing the baroclinicity of the atmosphere, affects the large-scale atmospheric circulation response to global warming[37,40,41], but the precise impacts are unknown since there is large inter-model spread in the projections of the NAWH anomaly and its spatial extent[38]. Even though reaching an AMOC tipping point with a consequent AMOC shutdown is deemed unlikely in projections of climate change of the next century[27,42], there is a wide range in the simulated AMOC decline rates[28,43–45]. Hence, the consequences of the inter-model spread in the AMOC response, including those on the NAWH, remain uncertain.

In this study we show that the inter-model spread in the AMOC response represents a major source of uncertainty in climate model projections of future changes in surface temperature, precipitation, and large-scale wind circulation. Specifically,

in models in which the AMOC declines more, there is on average reduced warming and precipitation increase, a southward shift of the ITCZ, and a poleward displacement of the zonal mean upper-level jet stream. Instead, in models in which the AMOC declines less, there is on average enhanced warming in the North Atlantic, a local increase in the hydrological cycle, and little displacement of the zonal mean upper-level jet stream.

## Results

**Inter-model spread in the AMOC response.** We examine an ensemble of 30 abrupt-4xCO$_2$ simulations (Supplementary Table 1) from the Coupled Model Intercomparison Project phase 5 ("CMIP5")[46] and phase 6 ("CMIP6")[47]. The AMOC strength anomalies at 26.5°N in the abrupt-4xCO$_2$ experiments are shown for 12 CMIP5 models in Fig. 1a and 18 CMIP6 models in Fig. 1b. All models show a decline from the preindustrial control climate. We note that the inter-model range is larger in CMIP6, although this could be influenced by the larger number of modeling centers contributing to the CMIP6 archive. In CMIP6, there are also some models that exhibit an overall stronger AMOC decline. The AMOC decline in CMIP5 ranges between about −4 Sv and −10 Sv (−27% to −58%), while in CMIP6 it ranges between about −1.5 Sv and −17.5 Sv (−18% to −74%). Supplementary Table 1 further provides statistics for each model. From here, we can tell that the majority of the models shows no abrupt shutdowns of the AMOC, although a collapse of the AMOC may be an overlooked possibility in state-of-the-art climate models[42]. On the contrary, we note that in some models the AMOC starts to recover towards the end of the abrupt-4xCO$_2$ simulation.

To investigate the differences in the AMOC decline across the models, we divide the CMIP5 + CMIP6 ensemble in two groups: the average of the ten models with the largest AMOC declines is referred to as the "large AMOC decline" group, while the average of the ten models with the smallest AMOC declines is referred to as the "small AMOC decline" group (Supplementary Table 2). For each of the models, we calculate the changes in the abrupt-4xCO$_2$ simulations from the preindustrial control mean climate. Table 1 shows that differences between the mean values of AMOC diagnostics are statistically different between the two groups, whereas Fig. 1c, d show the changes in the North Atlantic transects of the meridional overturning stream-function in the two groups. The transects show that the decrease in the AMOC tends to be more pronounced between 25°N and 40°N in both groups. We note that models with stronger AMOC in the mean climate generally belong to the large AMOC decline group, while models with weaker AMOC in the mean climate belong to the small AMOC decline group (indicated by contours in Fig. 1c, d), which is consistent with previous findings (see also Supplementary Fig. 1)[28,43].

While an inter-model spread in the AMOC response to climate change has been recognized before[27,28,44], here we investigate whether this spread leads to significant differences in global climate impacts. In order to separate the effect of the AMOC response from other processes, in the following we normalize the models dividing changes in the variables by the respective change in Global mean Surface Air Temperature (ΔGSAT). This reduces the influence of the inter-model spread due to other processes, and shows the expected change per degree of global warming, thus facilitating comparisons with other climate change simulations. ΔGSAT itself is weakly dependent on the change in AMOC (ΔAMOC; Supplementary Fig. 2); however, since the spread is large and the Pearson's r correlation coefficient between ΔGSAT and ΔAMOC is only 0.30 (not statistically significant), we argue that other climate feedbacks are more important in determining ΔGSAT[27,48,49]. Finally, we note that since in these experiments

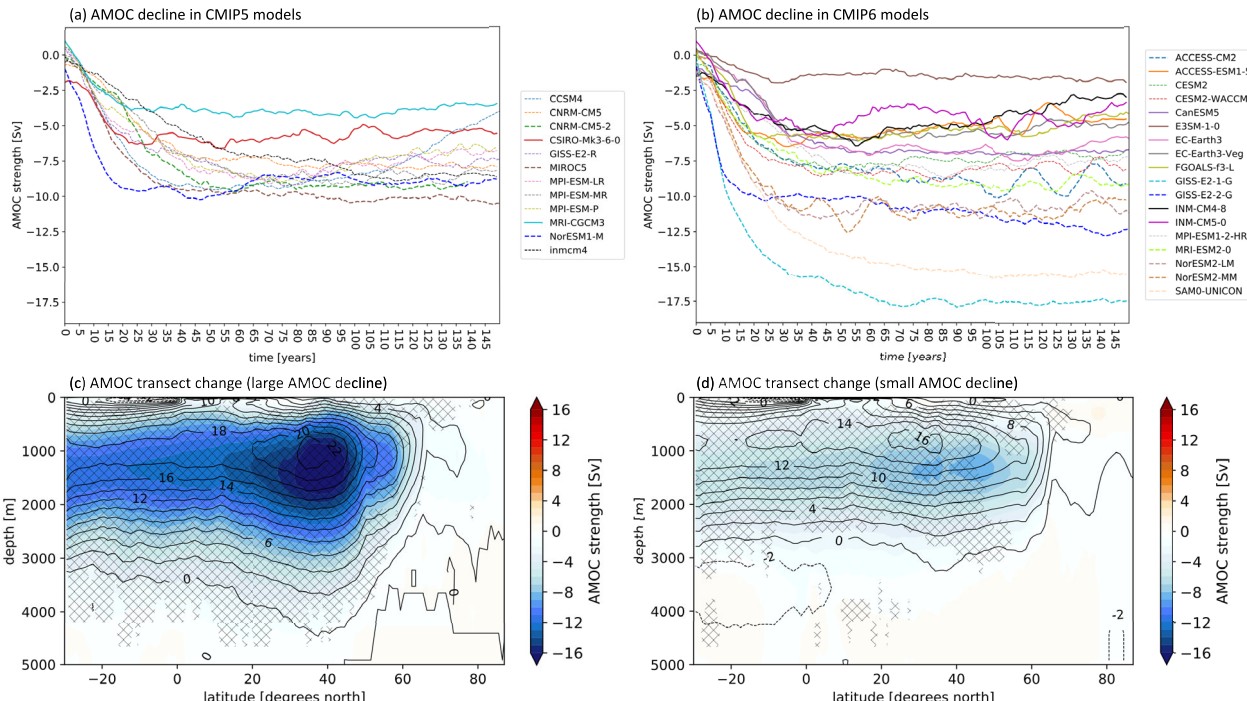

**Fig. 1 Inter-model differences in the AMOC response.** Panels **a**, **b** show the annual mean AMOC anomalies in the abrupt-4xCO$_2$ simulations at 26.5 °N for **a** CMIP5 and **b** CMIP6 models with respect to the mean AMOC strength computed from the preindustrial control. A 10-year running average smoothing has been applied to all curves for better visualization. Solid lines represent the models in the small AMOC decline group, while thick dashed lines represent the models in the large AMOC decline group. Models that do not belong to any groups, are represented by thin dashed lines. Panels **c**, **d** show transects of the ocean overturning stream-function in the North Atlantic for **c** the average of the large AMOC decline group, and **d** the average of the small AMOC decline group. Superimposed contours show the climatology computed from the preindustrial control of **c** the ten large AMOC decline models and **d** the 10 small AMOC decline models.

**Table 1 Differences in average values between the large and small AMOC decline groups.**

|  | Large AMOC decline | Small AMOC decline |
|---|---|---|
| Mean AMOC strength (preindustrial control) | 21.46 Sv | 16.82 Sv |
| Mean AMOC strength 4xCO$_2$ (years 90–139) | *10.13 Sv* | *11.98 Sv* |
| ΔAMOC | −11.32 Sv | −4.84 Sv |
| ΔAMOC (%) | −53.89 % | −29.33 % |
| ΔSPNA (ΔSST change in the SPNA) | 1.99 °C | 7.68 °C |
| ΔGSAT | 4.53 °C | 5.67 °C |
| ΔSPNA/ΔGSAT | 0.41 | 1.32 |

The differences between the means in the two columns are all statistically significant at the 90% level of a two-side Student's *t*-test, except for the values in italics. AMOC strength is computed at 26.5°N.

the AMOC is not artificially modified, but rather changes in response to the increase in CO$_2$, we are unable to fully separate the impacts of the AMOC from other forcings; hence, we rely on statistical tests to support our findings (see Methods).

**Surface temperature change**. Surface temperature change is shown in Fig. 2 for the large AMOC decline group (Fig. 2a) and small AMOC decline group (Fig. 2b). Figure 2a, b are the changes from the preindustrial control experiment, while Fig. 2c is their difference (large minus small AMOC change). The surface temperature at each grid point represents the normalized local temperature change per degree of global warming. Over the ocean, the surface temperature coincides with the SST, while over land it corresponds to the temperature at the surface. We interpret Fig. 2c as the expected impact of a larger AMOC decline in a future climate change scenario, compared to a smaller AMOC decline. In Fig. 2c, stippling indicates where the differences in the

simulated climate change between the small and large AMOC groups are statistically significant.

Figure 2a shows that in the models with the larger AMOC declines, there is a minimum SST warming in the North Atlantic, often referred to as the NAWH. Instead, Fig. 2b (small AMOC decline) shows no NAWH, but actually a relatively large SST warming in the SPNA. Figure 2c confirms that the change in surface temperature is drastically different between the two groups, especially in the North Atlantic and the Nordic Sea. In the Weddell and Ross seas in the Southern Ocean we see a reduced warming when the AMOC decline is larger (Fig. 2c), which, similarly to the North Atlantic, is associated with a larger decrease in the mixed layer depth (Supplementary Fig. 3). This suggests that in models in which the AMOC decline is larger, there is a stronger convection decrease also in the other deep-water formation regions associated with the global thermohaline circulation.

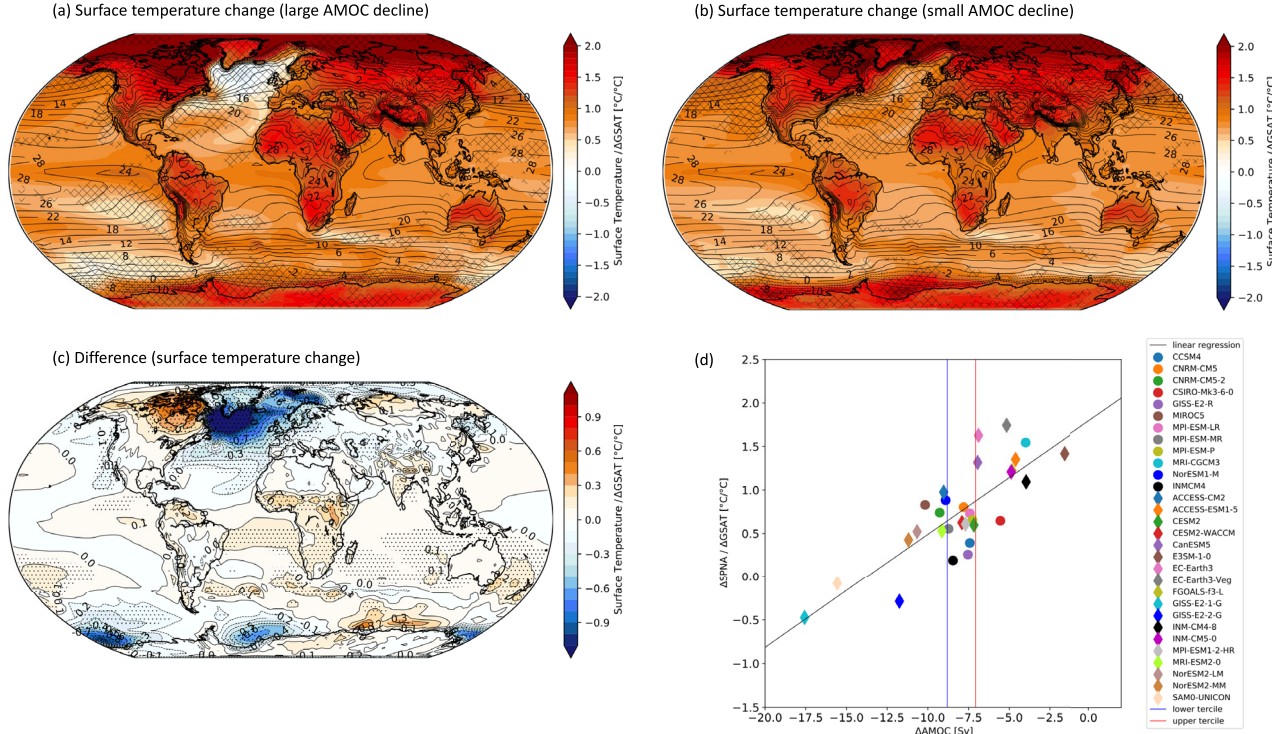

**Fig. 2 Surface temperature change associated with AMOC.** Panels **a**, **b** show the normalized annual mean surface temperature change in the abrupt-4xCO$_2$ with respect to the preindustrial control for **a** the average of the large AMOC decline group and **b** the average of the small AMOC decline group. Panel **c** is their difference (a minus b). For each model, surface temperature change is divided by the respective $\Delta$GSAT, hence units are °C per degree of global warming. In panels (**a**) and (**b**), superimposed contours show the climatological mean surface temperature computed from all models. In panel **c**, superimposed contours show the numerical values associated with the surface temperature differences of the colored contours. Panel **d** shows the change in subpolar North Atlantic SST ($\Delta$SPNA) divided by $\Delta$GSAT, against AMOC change ($\Delta$AMOC) in units of Sv. Circles represent CMIP5 models, while diamonds represent CMIP6 models. The dashed black line is the linear regression ($y = 0.13x + 1.8$) with $R^2$: 0.63. The blue and red vertical lines represent the lower and upper terciles of the $\Delta$AMOC distribution. Models to the left of the blue line belong to the large AMOC decline group, while models to the right of the red line belong to the small AMOC decline group.

In addition, the difference in near-surface air temperature change (not divided by $\Delta$GSAT) in the extra-tropical North Atlantic between the large and small AMOC decline groups (Supplementary Fig. 4) is very similar to the response to a disruption of the AMOC in water hosing model experiments[13–15]. Differently from the water hosing experiments, we do not see a relative warming in the Southern Ocean and in the Weddell and Ross seas, which is due to the fact that in the abrupt-4xCO$_2$ simulations the forcing is globally uniform and very different in nature from the freshwater anomaly added to the North Atlantic basin in the water hosing experiments. Nevertheless, despite the very different experimental designs, the similarity between these earlier experiments and our results further supports the hypothesis that the differences in the patterns of surface temperature change in the North Atlantic shown in Fig. 2 are largely influenced by the difference in AMOC decline between the large and the small AMOC groups, rather than by other processes.

Another interesting result is that a decline in the AMOC has a cooling effect on Europe, but not so much on North America (Fig. 2c and Supplementary Fig. 4c), which partly contradicts a previous study that argued that the effect of the AMOC on surface temperature in the mean climate is zonally uniform in the northern hemisphere[50]. In addition, the AMOC slowdown is associated with reduced Arctic amplification in the large AMOC decline group (Supplementary Fig. 5). Finally, we note that if instead of choosing the top and bottom ten models based on the AMOC decline, we simply split the models in half, above and

below the median AMOC decline, we still obtain the same temperature change pattern, although there is an overall smaller statistical significance. We also obtain similar results if we repeat this analysis only in the CMIP5 (or CMIP6) archive.

Figure 2d shows the SPNA SST change (area average of 50–70° N and 80°W–10°E) divided by $\Delta$GSAT, against the respective AMOC change for each of the 30 models in the CMIP5 + CMIP6 ensemble. This scatterplot shows that the relationship between the normalized SPNA SST change and AMOC change is robust across all models, and linear: the larger the AMOC decline, the smaller the projected temperature increase in the SPNA, and vice versa. The Pearson's $r$ correlation coefficient is 0.80 and is statistically significant. We interpret this relationship as the consequence of both a greater reduction in ocean heat transport in the North Atlantic and a larger heat uptake in the models in which the decline in AMOC is relatively stronger, which act to locally slow down the temperature increase. In addition, Fig. 2d shows that the influence of AMOC on surface temperature is not limited to the models with the largest and smallest AMOC declines, but applies to all models. Given this strong relationship, we argue that the inter-model spread in the AMOC response is a major source of uncertainty in SPNA SST change.

In summary, the AMOC decline acts as a regional negative feedback on temperature warming; however, this effect is much larger in the models featuring a large AMOC decline (compare Fig. 2a with 2b). This means that, based on whether the AMOC decline is large or small, there are drastically different changes in North Atlantic SST in the models. Table 1 further shows that the

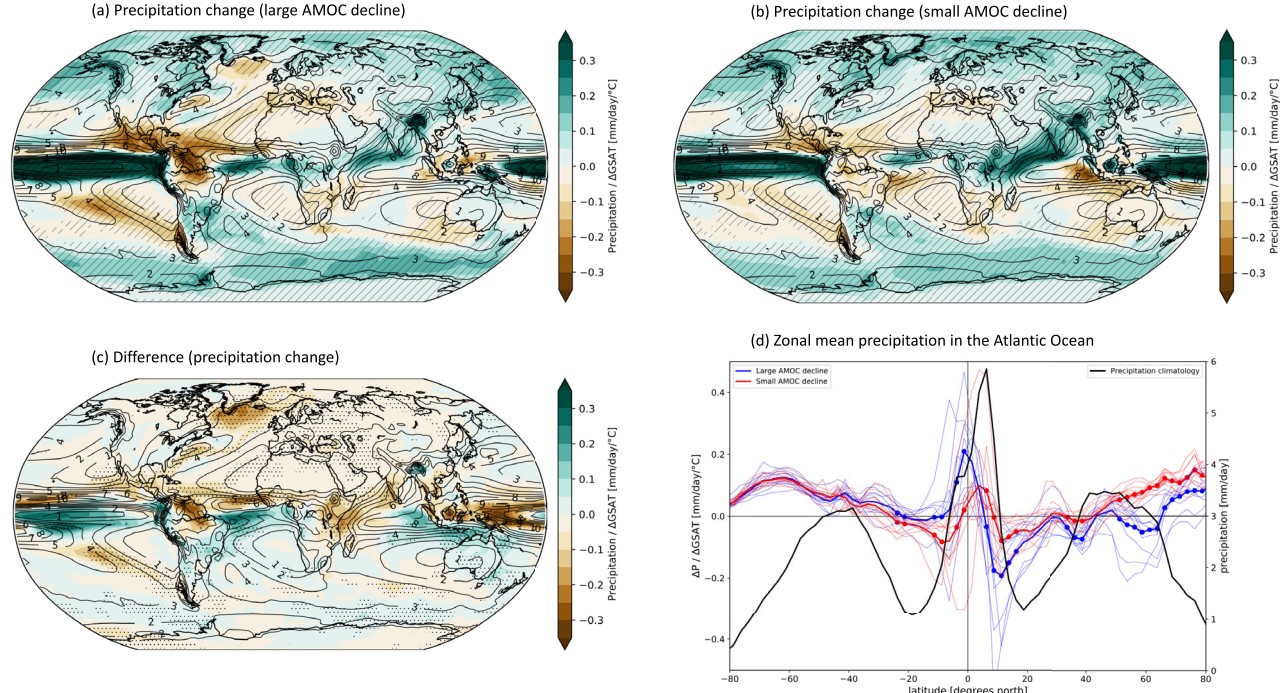

**Fig. 3 Precipitation change associated with AMOC.** Panels **a**, **b** show the normalized annual mean precipitation change in the abrupt-$4xCO_2$ with respect to the preindustrial control for **a** the average of the large AMOC decline group and **b** the average of the small AMOC decline group. Panel **c** is their difference (a minus b). For each model, precipitation change ($\Delta P$) is divided by the respective $\Delta GSAT$, hence the units are mm/day per degree of global warming. In panels **a**–**c**, superimposed contours show the climatological mean precipitation computed from all models. Panel **d** shows the zonal mean precipitation change in the North Atlantic sector in the (blue) large AMOC decline group and (red) small AMOC decline group. Thick lines are the group averages, while thin lines show each group member. Round markers indicate where the Student's $t$-test between the means of the two groups is significant at the 90% level. The black line is the zonal mean precipitation climatology computed from the preindustrial control of all models: units are of mm/day and the corresponding scale is located on right-side y-axis.

differences of the means of SPNA SST change and $\Delta GSAT$ between the two groups are statistically significant. In the following, we investigate whether the inter-model spread in the AMOC response may affect other aspects of climate change, including precipitation and large-scale atmospheric circulation.

**Precipitation change**. Figure 3 shows precipitation change divided by $\Delta GSAT$ (units of mm/day/°C) in the large AMOC decline group (Fig. 3a), small AMOC decline group (Fig. 3b), and their difference (Fig. 3c). In Fig. 3a–c, the ensemble mean of the precipitation climatology computed from all 30 CMIP5 + CMIP6 models is superimposed in contours. Similar to Fig. 2, Fig. 3a, b show the changes in the abrupt-$4xCO_2$ simulations from the preindustrial control. In Fig. 3c, stippling indicates where the differences in the simulated climate change between the large and small AMOC groups are statistically significant.

Figure 3a shows interesting dissimilarities from 3b. Generally speaking, the small AMOC decline group features the precipitation changes that we expect from the wet-get-wetter, dry-get-drier paradigm[51] (Fig. 3b), according to which precipitation over the ocean will increase over wet regions and decrease over dry regions. In contrast, the large AMOC group deviates from this paradigm (Fig. 3a). For example, over the North Atlantic mid-latitudes, the precipitation is projected to decrease over the Gulf Stream and over the SPNA in the large AMOC group (Fig. 3a), in stark contrast with the small AMOC decline group where the precipitation anomaly is of opposite sign and is actually projected to increase over those regions (Fig. 3b). Reduced rainfall over the SPNA is expected from an abrupt decline in the AMOC, and has been associated with a reduced evaporation from the ocean and a decrease in eddy moisture transport[17,52,53]. These same

mechanisms may operate in response to $4xCO_2$, but seem to prevail only in the models featuring a large AMOC decline. Overall, in the large AMOC decline group, precipitation changes driven by dynamic changes in atmospheric circulation seem to dominate over the thermodynamic changes related to the increase in specific humidity due to global warming[54–56].

In the Pacific Ocean, in the large AMOC decline group, there appears to be a more pronounced El Niño like response, with precipitation increasing in the eastern side of the equatorial Pacific and decreasing in the western side. This is associated with a larger warming in SST in the eastern side in the large AMOC decline group than in the small AMOC decline group (Fig. 2c). While these differences in precipitation and SST have poor statistical significance, negative SST anomalies associated with the Atlantic Multidecadal Variability, and positive SST anomalies in the eastern equatorial Pacific (and vice versa) have previously been linked[57], and this relationship is consistent with our results.

Focusing on the Atlantic Ocean, in Fig. 3d we compute the normalized zonal mean precipitation change in the large AMOC decline group (blue) and small AMOC decline group (red). The round markers on the blue and red lines indicate where the difference between the means of the two groups is statistically significant. For reference, the zonal mean precipitation climatology computed from all 30 CMIP5 + CMIP6 models is plotted in black. Some inter-model spread exists, but the differences between the groups are clear here: while, compared to the climatology (black), the small AMOC decline group (red) exhibits the wet-get-wetter and dry-get-drier behavior with increase in precipitation at the equator, decrease in the subtropics, and increase in the northern hemisphere mid-latitudes, the large AMOC decline group shows a different response. In the large

AMOC decline group, the peak of precipitation in the tropics (ITCZ), which normally sits north of the equator in the northern hemisphere (black), locally decreases and the anomaly shifts to the southern hemisphere. The shift of the ITCZ is associated with a reduced Arctic amplification when the AMOC decline is larger (Supplementary Fig. 5), which corresponds to a larger equator to pole temperature gradient compared to the small AMOC decline group.

In the mid-latitudes, we note that where the precipitation change is positive in the small AMOC decline group, it is of opposite sign in the large AMOC decline group (Fig. 3d). This, as noted above, could be related to a difference in the relative influence of thermodynamic and dynamic drivers of precipitation change, which are associated with the difference in warming in the North Atlantic between the two groups. If we examine the zonal mean ITCZ change globally, we still find a southward shift into the southern hemisphere, but this is dominated by the Atlantic contribution (Fig. 3c) and it is not statistically significant at the global level.

The Indian monsoon is also affected. While in both groups precipitation increases (Fig. 3a, b), in the large AMOC decline group precipitation does not increase as much as in the small AMOC group (Fig. 3c). This suggests that the AMOC may modulate the response of the Indian monsoon to climate change, with important societal implications. A connection between the AMOC, the warming of the Indian Ocean and the summer monsoon has been noted before both in climate change scenarios[58] and on inter-decadal variability[59], and has been associated with the north-south temperature gradient across Eurasia.

**Atmospheric circulation change**. Given that the NAWH influences the north-south gradient of air temperature in the northern hemisphere (Supplementary Fig. 5), which may affect the mid-latitude jet, we investigate whether there are any significant differences in the response of the mid-latitude westerly winds in relation to the AMOC decline. Figure 4 shows the normalized zonal mean wind speed change in the abrupt-4xCO2 simulations from the preindustrial control mean in boreal winter (DJF) for the large AMOC decline group (Fig. 4a), small AMOC decline group (Fig. 4b), and their difference (Fig. 4c). For each model, the change in zonal mean wind speed is divided by $\Delta$GSAT (units of m/s/°C). The difference shown in Fig. 4c, similarly to the other figures, is attributed to the AMOC response difference between the two groups. The climatological mean computed from all models is superimposed in contours in Figs. 4a–c, while stippling in Fig. 4c indicates statistical significance. By definition, climatological positive values indicate westerly wind speed, while negative values indicate easterly wind speed.

The patterns of change are similar in the large and small AMOC groups: however, their difference (Fig. 4c) reveals that in the northern hemisphere mid-latitudes there is a statistically significant increase in westerly wind speed poleward of the climatological maximum, and a decrease to the south. From a purely thermodynamic standpoint, in response to climate change there is a tug of war between the contrasting effects on the jet of Arctic amplification and tropical upper troposphere heating[39,60,61]. While Arctic amplification would, on its own, push the mid-latitude jet closer to the equator, the tropical heating together with the expansion of the Hadley cell would push the jet poleward[62]. Figure 4c shows that the effect of the tropical heating seems to be stronger when there is larger AMOC decline because in these models Arctic amplification is on average reduced compared to the models in which there is a smaller AMOC decline (Supplementary Fig. 5).

Further examination reveals that this mechanism explains the changes in the thermally driven upper-level jet, but not the low-level eddy driven jet. Figure 4d shows, for each model, the change in the latitude of the maximum westerly wind speed at 250 hPa divided by $\Delta$GSAT, against the AMOC change. There is a correlation between the amplitude of the AMOC decline and the northward displacement of the jet: the stronger the AMOC decline, the more the jet is displaced poleward. In contrast, in models where the AMOC decline is smaller, there is actually an equatorward displacement. The Pearson's r correlation coefficient is −0.58 and it is statistically significant; however, when we exclude two models (SAM0-UNICON and GISS-E2-1-G) that seem to behave differently from the ensemble, the Pearson's r correlation coefficient increases to −0.70. Although statistically significant, this relationship is weaker compared to the relationship between the NAWH and the AMOC response (correlation coefficient of 0.80). We also do not find a similarly robust relationship between the eddy-driven jet at 850 hPa and AMOC change, which could be related to the fact that the NAWH has been found to have a relatively weak impact on the eddy-driven jet, compared to other drivers[39,60].

In the southern hemisphere, we see an intensification of the westerlies that is more pronounced in the large AMOC decline group (Fig. 4c). A mechanism proposed in a previous study[63] could explain this difference: the more southward dislocation of the Hadley cell in the large AMOC decline group (Fig. 3c) could contribute to further strengthen the southern hemisphere westerlies through the weakening of the subtropical jet. However, a more detailed analysis of the interaction of the eddies with the mean flow is needed to corroborate this idea.

## Discussion

The AMOC is expected to decline in response to increasing greenhouse gas concentrations;[27,28] however, the role of the AMOC decline in future climate change is unclear. Several studies have examined the impacts of an AMOC shutdown in idealized model experiments where the AMOC is artificially halted[13–15], and a few of them have examined the impacts of the AMOC decline in the context of future climate change[17,32–34]. However, none of them has approached this problem in a multi-model context using state-of-the-art global climate models. In this study, we address this question specifically examining how the inter-model range in the AMOC decline affects projected climate change in response to an abrupt quadrupling of CO2 in a suite of 30 climate models participating in the CMIP5 and CMIP6 archives.

The reduced northward heat transport by the AMOC acts as a negative feedback on SST warming in the North Atlantic, which results in a minimum warming in the North Atlantic in the models with stronger AMOC decline. However, in models with weaker AMOC decline, this effect is small and, on average, there is actually a relatively larger warming in the SPNA. We find that these drastically different SST warming scenarios are associated with large-scale impacts on precipitation and mid-latitude circulation responses. In the models with larger AMOC decline, the precipitation over the oceans does not follow the wet-get-wetter, dry-get-drier paradigm: in fact, in the Atlantic there is a southward shift of the zonal mean ITCZ, and a reduction in precipitation over the Gulf Stream and the SPNA. In addition, the mid-latitude thermally driven jet strengthens and moves poleward. In contrast, in the models with smaller AMOC decline, the precipitation response is as expected by the wet-get-wetter, dry-get-drier paradigm, while the jet's displacement is either small or equatorward. The differences in the response of large-scale atmospheric circulation and precipitation are partly explained by

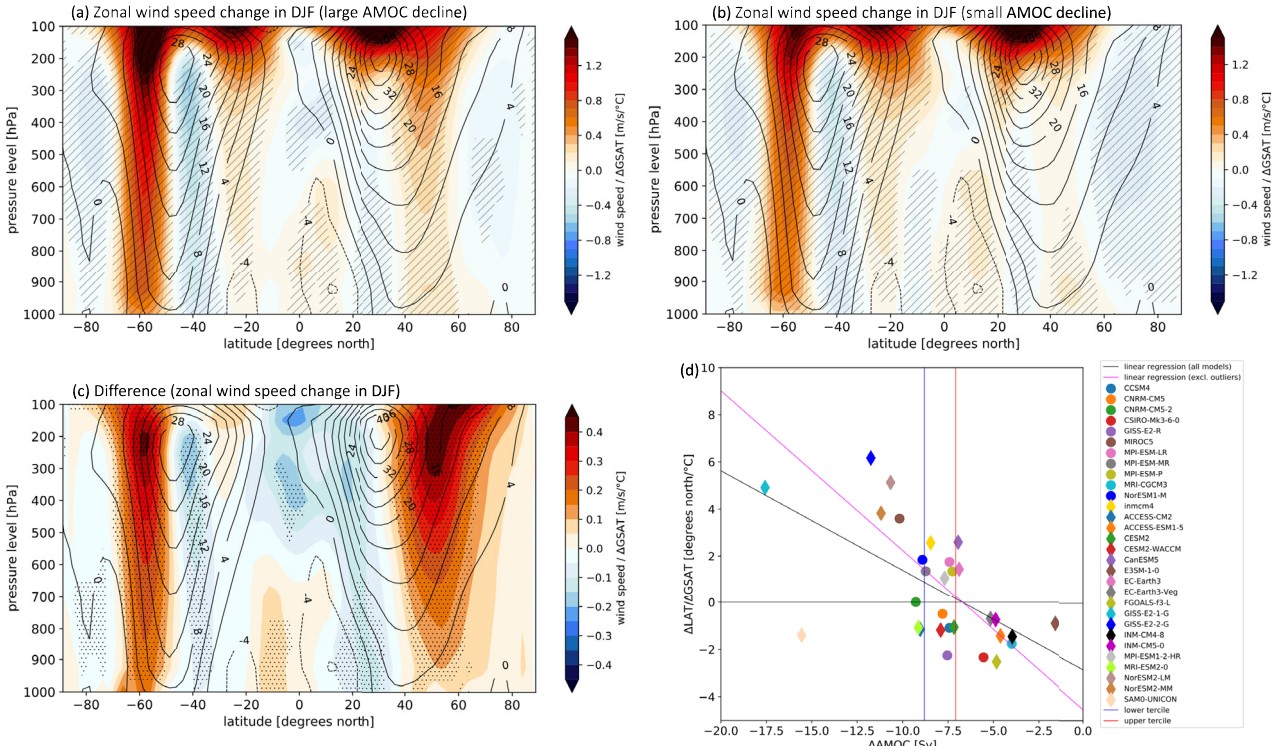

**Fig. 4 Winter wind speed change associated with AMOC.** Panels **a**, **b** show the normalized zonal mean wind speed change in DJF in the abrupt-4xCO$_2$ with respect to the preindustrial control for **a** the average of the large AMOC decline group and **b** the average of the small AMOC decline group. Panel **c** is their difference (a minus b). For each model, the zonal mean wind speed change is divided by the respective ΔGSAT, hence units are of m/s per degree of global warming. In panels **a**–**c**, superimposed contours show the climatological zonal mean wind speed in DJF computed from all models. Panel **d** shows the change in the latitude of the maximum westerly wind speed at 250 hPa in the northern hemisphere (ΔLAT) divided by ΔGSAT, against AMOC change (ΔAMOC). Circles represent CMIP5 models, while diamonds represent CMIP6 models. The dashed black line is the linear regression including all models ($y = -0.42x - 2.84$ with $R^2$: 0.33), while the dashed magenta line is the linear regression excluding the models GISS-E2-1-G and SAM0-UNICON ($y = -0.68x - 4.54$ with $R^2$: 0.48). The blue and red vertical lines represent the lower and upper terciles of the ΔAMOC distribution. Models to the left of the blue line belong to the large AMOC decline group, while models to the right of the red line belong to the small AMOC decline group.

differences in Arctic amplification. Even though Arctic amplification also depends on other factors[27], we find that models in which the AMOC decline is larger exhibit a reduced Arctic amplification.

Our results are in agreement with, and extend, the findings of a recent study[17], in which an idealized model experiment was run with one global climate model to examine the impacts of the AMOC decline in future climate change. We also find that the differences in climate change over the North Atlantic between the large and small AMOC decline groups, which we interpret as related to the AMOC response, are similar to the climate change patterns expected by a shut down of the AMOC, which have been largely investigated in single model water hosing simulations[13–15]. However, our results further suggest that more caution should be used when investigating mechanisms of climate change related to precipitation and large-scale atmospheric circulation patterns. In fact, it seems that some of the uncertainties in precipitation and the jet stream, which were previously attributed to dynamic changes in wind circulation[54,55], are also influenced by the inter-model spread in the AMOC response and its impact on the NAWH[39].

The main caveat of our study is that it remains difficult to fully separate the influence of the AMOC from other processes using the existing experiments in the CMIP5 and CMIP6 archives. Future work should focus on refining a method to better isolate the AMOC from other feedback mechanisms, which could include ad hoc simulations and a coordinated model intercomparison. Having here laid the foundation that the AMOC

response is related to specific impacts in projections of future climate change, additional work is needed to mechanistically explain the links between the AMOC, the surface temperature change, and the associated effects on precipitation and atmospheric circulations.

In conclusion, the key finding of this work is that the inter-model spread in the AMOC response is linked to the uncertainty in the projections of a number of societally important atmospheric variables. The implication is that not only is ocean circulation important for climate change, but also that the uncertainty in the AMOC response may amplify the inter-model spread in the projections of temperature, precipitation, and large-scale wind circulation. We note that there has been recent progress in showing that model biases in the simulation of the mean climate are linked to the amplitude of the AMOC response[64]. We also have found a dependence of the amplitude of the AMOC decline on the mean strength of the AMOC, which corroborates previous results[28,43]. This suggests that continued observational efforts in the North Atlantic can help constrain the simulation of the mean climate, thereby reducing the uncertainty in projections of future climate change.

## Methods
**Data**. In this study, we examine the preindustrial control and abrupt-4xCO$_2$ experiments from the CMIP5 and CMIP6 archives. In the preindustrial control experiment, the atmospheric concentration of CO$_2$ is held fixed at ~284 ppm and the model variability is entirely driven by internal processes. In the abrupt-4xCO$_2$ experiment the concentration of CO$_2$ is suddenly increased to four times the value of the preindustrial control experiment and held at this value throughout the experiment.

We use the model output of 12 CMIP5 and 18 CMIP6 models. Supplementary Table 1 lists the models used in this study, and shows relevant statistics.

**Analysis**. All datasets are interpolated to a common $2.5° × 2.5°$ grid before the analysis. We use only one ensemble member for each model (r1i1p1 for CMIP5 and r1i1p1f1 for CMIP6), and we find no significant differences when additional ensemble members are included for the models that made them available. For the EC-Earth3 model, the r1i1p1f1 ensemble member for the abrupt-4xCO2 simulation was not available when we accessed the data, hence we use r3i1p1f1.

We calculate the AMOC index as the maximum of the ocean meridional overturning stream-function at 26.5°N in the Atlantic Ocean for each year (results are similar using 40°N). We note that for the model FGOALS-f3-L we were unable to access the ocean meridional stream-function from the preindustrial control, hence for this model we use the first year of the abrupt-4xCO2 simulation to compute the mean AMOC strength. This choice is motivated by the fact that for all the other models there is very good agreement between the value of the AMOC strength computed from the first year of the abrupt-4xCO2 simulation and the mean of the preindustrial control (Supplementary Fig. 6): the Pearson's $r$ correlation coefficient is 0.98. For reference, Supplementary Table 1 reports the mean AMOC strength computed from the preindustrial control experiments, the AMOC index change, and other statistics.

To quantify the influence of the inter-model spread in the AMOC response, we first calculate the impact of the quadrupling of $CO_2$ as the difference between the mean of the years 90 through 139 (total of 50 years) in the abrupt-4xCO2 simulation and the mean of the years 50 through 199 (total of 250 years) in the preindustrial control simulation for each variable and all models. Choosing other time frames from the preindustrial control to compute the differences leads to similar results. We decided to use the years 90 to 139 for the abrupt-4xCO2 simulations because after year 90 all the models reach a plateau in the AMOC response (Fig. 1). We use the year 139 as the final year because we want to maximize the number of models available to analyze, and one of the models only provides 140 years instead of 150. Moreover, some models show a recovery of the AMOC towards the end of the abrupt-4xCO2 simulation (Fig. 1), thus excluding the last 10 years limits this influence.

To specifically investigate the role of the AMOC in driving climate changes relative to other processes, we form two groups of models based on the AMOC index change: the large AMOC decline group includes the ten models with the largest AMOC declines, while the small AMOC decline group includes the ten models with the smallest AMOC declines (Supplementary Table 2). We interpret the difference between these two groups (large AMOC decline minus small AMOC decline) as the effect of the AMOC response on the differences in the simulated climate change impacts between the two groups. For each model, we also divide the changes in each variable by the respective change in ΔGSAT to reduce the influence of other processes and feedbacks. ΔGSAT is computed as the area averaged global mean near-surface air temperature change from the preindustrial control experiments. If instead of dividing into groups based on the absolute value of AMOC strength in units of Sv, we divide based on the percent AMOC change from the preindustrial control, we find similar results. An alternative method to identify the role of the AMOC is to perform linear regressions of each variable on the AMOC anomalies; however, this approach masks the distinctive patterns (e.g., precipitation in Fig. 3) that we want to highlight.

**Statistical tests**. To investigate whether the changes associated with the AMOC response are statistically significant in the spatial maps, we perform a two-tailed $t$-test on the differences between the large and small AMOC decline groups, assuming equal variance in the two groups. Where this test indicates that results are statistically significant at the 90% level, we argue that the differences are driven by the different AMOC responses in the two sets of models. This is indicated by the squared stippling in the figures (Figs. 2c, 3c, 4c, Supplementary Fig. 4c, and Supplementary Fig. 5c). We further assess statistical significance by forming two groups of 500 samples of randomly picked groups of ten models among the 30 CMIP5 + CMIP6 models, without repetition. We perform a $t$-test and a $z$-test to check whether the difference between large and small AMOC decline groups is statistically different from the 500 differences of the two randomly chosen groups. The results of the random tests show even better significance for all variables than the two-tailed $t$-test, hence they are not shown.

We also provide measures of inter-model reliability to assess whether within the large and small AMOC decline groups there is good inter-model agreement in the changes from the preindustrial control climate. We use two definitions. In panels (a) and (b) of Figs. 3 and 4, reliability is indicated by the "/" hatches and is defined where at least 80% of the models in each group agree in sign with the ensemble mean of the group for each grid point. This definition is informative when the expected change could be either positive or negative. However, for other variables, such as surface temperature, the change in response to 4xCO2 forcing is almost always of the same sign for all the models. For this reason, in panels (a) and (b) of Figs. 1, 2, Supplementary Fig. 4, and Supplementary Fig. 5, we use a more stringent definition of reliability, indicated with the "x" hatches. In this case, the hatched areas show where at least a certain percentage of models fall on either side of the median of all the CMIP5 + CMIP6 models (i.e., whether the majority of models is above or below the median change). The specific threshold for the "x" hatches

depends on the variable: for Fig. 2, Supplementary Fig. 4, and Supplementary Fig. 5 is 60%, while for Fig. 1 is 80%.

For all of the scatterplots, we define the Pearson's $r$ correlation coefficients as statistically significant when the $p$ values derived from the two-tailed Student's $t$-distribution exceed a 95% probability threshold. For Fig. 4d, to test whether any of the models affects the validity of the statistical test, we perform an additional test, in which we leave one model out at the time, and calculate the correlation coefficient across the remaining models: even so, we find that all correlation coefficients are statistically significant, and that the average of them is −0.58, the same as when we exclude the two outliers in Fig. 4d (GISS-E2-1-G and SAM0-UNICON). In addition, if we exclude the two outliers from the averages in the map plots of Figs. 2 and 3 for surface temperature and precipitation, we still find very similar results.

## Data availability
The model output data that support the findings of this study are publicly available through the CMIP5 and CMIP6 Earth-system Grid Federation archives, and can be accessed at https://esgf-data.dkrz.de/search/cmip5-dkrz/ and https://esgf-data.dkrz.de/search/cmip6-dkrz/.

## Code availability
All codes used for the analysis of the data for this work are available upon request from K.B.

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

## Acknowledgements

We thank Virna Meccia, Paolo Davini, and Mark Zelinka for helpful discussions. This project has received funding from the European Union's Horizon 2020 research and innovation program under grant agreements No. 641816 (CRESCENDO) and No. 820970 (TiPES). This is TiPES contribution #55.

## Author contributions

K.B., M.A., J.v.H., and S.C. conceived the project. J.v.H and K.B. retrieved the data. K.B. analyzed the data and wrote the manuscript. K.B., M.A., J.v.H., and S.C. participated in the discussion and analysis of the results, and edited the manuscript.

## Competing interests

The authors declare no competing interests.
