## [Peer Review File · Nature Communications]

REVIEWER COMMENTS

Reviewer #1 (Remarks to the Author):

Summary Statement

This paper argues that under CO₂ forcing, model spread in surface climate changes are related to spread in AMOC declines. The paper does this by analyzing 4xCO₂ simulations in the CMIP5/CMIP6 archive and dividing these simulations into large and small AMOC decline groups. The assumption is that the differences between the two model groups are driven by the differences in the AMOC decline. This assumption needs to be more carefully justified, in particular in particular related to the claim that the AMOC changes are leading to changes in the winds (which seems to be the only "new" result of this work, as the impact of AMOC changes on the North Atlantic warming hole and ITCZ shifts has been already extensively documented).

Range of AMOC declines

The paper claims that the range of AMOC declines in CMIP6 models is larger than in CMIP5 models, but this is likely largely explained by the fact that there are more CMIP6 models included from a larger range of modeling centers. For example, in CMIP6 the smallest AMOC decline (E3SM-1-0) and the second largest AMOC decline (SAMO-UNICON) come from modeling centers for which CMIP5 models are not included. There seems to be little change when comparing the new generation of models from the same modeling center, with the exception of the GISS model, GISS-E2-1-G. Can this greater sensitivity be explained by a change in the model (one that does not seem to also be present in GISS-E1-1-G)?

Relationship between AMOC declines and surface climate

The analysis assumes that differences in surface climate changes between the large and small AMOC decline runs are related to the difference in changes in the AMOC. This assumption is plausible, and the fact that the changes in SST and precipitation are similar to those that have been related to the AMOC in prior work is comforting (although this also mean that much of the AMOC impacts described in this work are not new). The precipitation changes in the Pacific were not described very clearly (and they do not appear to be statistically significant anyhow?). The relationship between AMOC declines and the strength of the zonal winds (Figure 4, and a "new" aspect of this paper) is much less convincing. For example, the linear relationship between the AMOC changes and those of the latitude of the maximum wind speed is quite weak ($R^2=0.33$, Figure 4d) and it quite clearly depends on a single model (GISS-E2-1-G). I do not believe that computing the linear fit and R^2 when removing two "outliers" (models that do not fit your linear hypothesis) is appropriate.

For the plots of precipitation and the easterly wind, in panels a and b I would like to see stippling using the more stringent (and relevant in this case) test that 80% of the models agree on the sign of the change relative to that of the ensemble mean (as was done for SST). Since we are making conclusions about the difference between 2 groups of models, how these models are different from the ensemble mean is the appropriate measure.

I have also included some comments on the attached pdf. No need to respond to these comments separately as the important points are detailed in my written review.

Reviewer #2 (Remarks to the Author):

Review on "Future climate change scenarios shaped by inter-model differences in Atlantic Meridional Overturning Circulation response" by Bellomo et al.

The authors study global climate impacts of varying degrees of AMOC slowdown under external climate forcing. The methodology of contrasting strong vs. weak AMOC decline in CMIP5 and CMIP6 models is somewhat novel, even though the conclusions are not too surprising. I think this study would advance our understanding on how AMOC might regulate global climate response to external climate forcing. I would recommend its acceptance for publication once the specific comments (see below) are addressed.

Specific Comments

Line 129-131, "Fig. 2c confirms that the change in surface temperature is drastically different between the two groups, especially in the North Atlantic and the Arctic, but also in the Southern Ocean in the areas of deep-water formation. ":

Besides what you have described, two other features on Fig. 2c seem interesting:

1) The contrast of warming and cooling anomalies, respectively, over the North America Continent (NAC) and the subpolar North Atlantic Ocean (SPNA). What mechanism do you think may have caused this pattern? All else being equal, if the NAC warms up and cold air outbreaks from the continent weakens, AMOC would decline in response to the resulting weakened surface buoyancy loss in the SPNA, which then leads to the warming hole in the SPNA. This potentially could explain the land-sea contrast seen in Fig. 2c, but in this case, the ultimate driver of the pattern in Fig. 2c is not AMOC but surface buoyancy anomaly in the SPNA. Do you think this might be what's going on here?

2) The cold anomalies in the Weddell Sea and Ross Sea in the Southern Ocean under stronger AMOC declines. AMOC decline is often associated with bi-polar adjustment in the SST/surface air temperature, that is, warming of the SO and cooling of the NH polar region when AMOC weakens. This picture would not fit with what's shown in Fig. 2c, where the zonal mean surface air temperature anomaly in the SO is negative when AMOC weakens more. What's the reason(s) behind the cold anomalies in the SO?

Minor comments

Line 359-361, "which is defined where at least a certain percentage of models in each group agree on the sign of the difference between each model's change and the median change of all CMIP5+CMIP6 models.":

At first, I was confused by this statement. It might be helpful to state that this method identifies if the majority of models in the large or small AMOC decline group fall on one side of the median of all the models.

Clarification

Line 643. " The '*' stippling indicates..."; Line 644-646, "Line In panels (a), (b) and (c), superimposed contours show the climatological mean surface temperature computed from all models." :

Fig. 2c stippling is not * but squares. Contours in Fig. 2c are not climatological mean surface temperature - it's the same as the color shaded field.

Reviewer #3 (Remarks to the Author):

This is one of the most interesting CMIP6 analyses I've seen and I strongly recommend publication. The authors have found that changes in the Atlantic meridional overturning circulation determine a wide range of climate responses associated with global warming. There are strong connections with patterns of surface temperature change, precipitation patterns and zonal winds. This has wide implications on a variety of fields: oceanographers who need to characterize the observed AMOC and its changes; modelers who need to reproduce the observed circulation with more fidelity; and climate impact scientists, who should recognize that precipitation across the globe is affected by the AMOC.

Many of the questions I had while reading were answered via the supplementary information or methods. I have just a few comments. First, it might be worth comparing results in the Southern Hemisphere with Shih-Yu Lee et al 2011 to see if the larger shift in the SH might be associated with that mechanism. Second, I

would probably recommend calling the winds westerlies instead of easterlies but there's no loss of generality there. Finally, I found the detailed pattern of precipitation quite fascinating, and if there's room in the text I might recommend a little more detailed description of the locations with north/south shifts vs other mechanisms.

To all: please note that we improved the quality of the figures (changed colormaps and projections).

Reviewer #1 (Remarks to the Author):

Summary Statement

This paper argues that under CO₂ forcing, model spread in surface climate changes are related to spread in AMOC declines. The paper does this by analyzing 4xCO₂ simulations in the CMIP5/CMIP6 archive and dividing these simulations into large and small AMOC decline groups. The assumption is that the differences between the two model groups are driven by the differences in the AMOC decline. This assumption needs to be more carefully justified, in particular in particular related to the claim that the AMOC changes are leading to changes in the winds (which seems to be the only “new” result of this work, as the impact of AMOC changes on the North Atlantic warming hole and ITCZ shifts has been already extensively documented).

We thank you for providing useful insights regarding our work, and pointing out some weaknesses. In the revised version of the manuscript, we addressed your specific comments below. In particular, we provided a better discussion of the innovative aspects of our research, which we believe are not limited to the changes in the winds (see modified text in ‘Discussion’ at L272-280). In fact, ours is the first study that points to the role of AMOC in generating inter-model uncertainty in future climate change scenarios in 3 key atmospheric variables: surface temperature, precipitation and wind speed, using a large ensemble of GCMs. Previous studies all used either single model experiments or water-hosing experiments in the absence of increasing CO₂, but didn’t examine the role of AMOC in a multi-model context. For what concerns the assumption of the role of the AMOC in driving these changes, our conclusions are supported by statistical tests as described in the Materials and Methods. Note that in response to one of your comments related to this, we improved the definition of inter-model reliability for precipitation and wind speed.

Range of AMOC declines

The paper claims that the range of AMOC declines in CMIP6 models is larger than in CMIP5 models, but this is likely largely explained by the fact that there are more CMIP6 models included from a larger range of modeling centers. For example, in CMIP6 the smallest AMOC decline (E3SM-1-0) and the second largest AMOC decline (SAMO-UNICON) come from modeling centers for which CMIP5 models are not included. There seems to be little change when comparing the new generation of models from the same modeling center, with the exception of the GISS model, GISS-E2-1-G. Can this greater sensitivity be explained by a change in the model (one that does not seem to also be present in GISS-E1-1-G)?

The figure below shows the ensemble including only models developed by the same modeling centers in CMIP5 (dashed lines) and CMIP6 (solid lines). Models are color-coded based on the modeling center. We don’t think we can draw a conclusion on whether the spread is larger in CMIP5 or CMIP6 without a more thorough comparison between CMIP5 and CMIP6, which is outside the scope of this paper. However, to address the issue that was raised here, we decided to remove the strong statement that was in the previous version of the manuscript: we changed the sentence starting at L83 to *‘We note that the inter-model range is larger in CMIP6, although this could be influenced by the larger number of modeling centers contributing to the CMIP6 archive.’*

For what concerns the GISS model, this much stronger AMOC decline in the GISS-E2-1-G compared to the GISS-E2-2-G version has been noticed before in the literature, but a definite answer has yet to be found. The following excerpt is quoted from Rind et al. 2020 and it's the most complete explanation we were able to find:

'While a detailed analysis of the different responses is beyond the scope of this paper, a key element relates to the deep water in the Labrador Sea and regions just to the east in the respective models. E2.1, with warmer than observed SSTs in that region (see section 3.2.1), provides dense water via excessive salinity. Apparently, this proves extremely susceptible to the increased warmth and precipitation in the higher CO₂ climate. E2.2, being a colder run, has colder air coming off the continent, and the contrast with ocean temperatures induces strong evaporation, resulting in a locally-derived increase in salinity; combined with the colder water in that control run, this seems to make the production more stable. A full analysis of this difference will be presented elsewhere.'

Paraphrasing, we understand this as being related to the mean climate in the two models. We further note that the difference between GISS-E2-1-G and GISS-E2-2-G is in the atmosphere (Rind et al. 2020): more specifically, the GISS-E2-2-G model is optimized for the middle atmosphere (i.e., there are more vertical layers). On the other hand, they share the same ocean model ('G', which stands for the 'GISS' ocean model). Since even in this very recent paper there doesn't seem to be a definite answer on this matter, we decided not to add any specific comment related to this in the revised manuscript.

Reference:

Rind, D., Orbe, C., Jonas, J., Nazarenko, L., Zhou, T., Kelley, M., ... Schmidt, G. (2020). *GISS Model E2.2: A Climate Model Optimized for the Middle Atmosphere. Model Structure, Climatology, Variability and Climate Sensitivity. Journal of Geophysical Research: Atmospheres.* doi:10.1029/2019jd032204

Relationship between AMOC declines and surface climate

The analysis assumes that differences in surface climate changes between the large and small AMOC decline runs are related to the difference in changes in the AMOC. This assumption is plausible, and the fact that the changes in SST and precipitation are similar to those that have been related to the AMOC in prior work is comforting (although this also mean that much of the AMOC impacts described in this work are not new).

In the revised ‘Introduction’ and ‘Discussion’ sections we make a stronger case for what is new in the paper. Please see an overview of what we have added in response to the ‘Summary Statement’ above.

The precipitation changes in the Pacific were not described very clearly (and they do not appear to be statistically significant anyhow?).

Much previous work has focused on the relative roles of the North Atlantic (AMOC) and the tropics (specifically changes in the Pacific) in driving drastic global climate changes, such as glacial/interglacial cycles. The discussion about which one of these two regions provides the largest contribution to abrupt climate changes is far from settled. In this general context, we feel it’s important to discuss the signal that we see even if it is small.

We already do use caution in the text at L199-206, specifically in this statement: ‘*While this difference has poor statistical significance, a negative correlation in which negative SST anomalies associated with the Atlantic Multidecadal Variability are linked with positive SST anomalies in the eastern equatorial Pacific (and vice versa), has been noted before⁵⁷, and is consistent with our results*’. We describe the signal as not as statistically significant as in other areas of the ocean, such as the Atlantic Ocean. Note that we also removed the sentence referring to uncertainties in future projections of ENSO from the revised manuscript (it was at L274-275 in the submitted manuscript).

However, without further investigation involving ad-hoc model analysis (or experiments) we are unable to disentangle the mechanisms at play in the equatorial Pacific, including the (dominant) local ocean-atmospheric dynamics. This further analysis is outside the scope of the present paper.

The relationship between AMOC declines and the strength of the zonal winds (Figure 4, and a “new” aspect of this paper) is much less convincing. For example, the linear relationship between the AMOC changes and those of the latitude of the maximum wind speed is quite weak ($R^2=0.33$, Figure 4d) and it quite clearly depends on a single model (GISS-E2-1-G). I do not believe that computing the linear fit and R^2 when removing two “outliers” (models that do not fit your linear hypothesis) is appropriate.

Related comment from marked up PDF at L241: ‘These are actually models with some of the largest AMOC declines. I don’t think that it is justified to remove these models, but if you are going to do this, you should show what happens if you exclude these models in your analysis of SST and precipitation. Furthermore, to me it looks like your linear relationship depends highly on one model, GISS-E1-1-G’

In addition to panel (d) in fig. 4, our original submission already provided a physical explanation and figures supporting why a larger AMOC decline would lead to a more poleward shift (namely, a reduced Arctic Amplification, fig. S4). However, to specifically address the concerns related to the significance of the regression lines in fig. 4d and the dependence on one model (GISS-E2-1-G), we performed an additional statistical test in which we left one model out at the time and computed the correlation coefficient (Pearson’s) and statistical significance for all the other models. We find, see table below, that while excluding GISS-E2-1-G gives the minimum correlation (-0.496) and excluding SAM0-UNICON gives the maximum correlation (-0.724), the average of all correlations (excluding one model at the time) is -0.577, which is the same as the value reported in the paper (-0.58) of the correlation of all models minus the aforementioned two. All correlations are statistically significant. In the table below, we include p-values (two-sided p-value of a Student’s t-distribution).

	model left out	corr. coeff.	pval	prob %					
0	CCSM4	-0.578	0.001	99.9	16	CanESM5	-0.596	0.001	99.9
1	CNRM-CM5	-0.579	0.001	99.9	17	E3SM-1-0	-0.580	0.001	99.9
2	CNRM-CM5-2	-0.583	0.001	99.9	18	EC-Earth3	-0.584	0.001	99.9
3	CSIRO-Mk3-6-0	-0.566	0.001	99.9	19	EC-Earth3-Veg	-0.572	0.001	99.9
4	GISS-E2-R	-0.586	0.001	99.9	20	FGOALS-f3-L	-0.559	0.002	99.8
5	MIROC5	-0.568	0.001	99.9	21	GISS-E2-1-G	-0.496	0.006	99.4
6	MPI-ESM-LR	-0.583	0.001	99.9	22	GISS-E2-2-G	-0.550	0.002	99.8
7	MPI-ESM-MR	-0.576	0.001	99.9	23	INM-CM4-8	-0.563	0.001	99.9
8	MPI-ESM-P	-0.582	0.001	99.9	24	INM-CM5-0	-0.572	0.001	99.9
9	MRI-CGCM3	-0.560	0.002	99.8	25	MPI-ESM1-2-HR	-0.579	0.001	99.9
10	NorESM1-M	-0.576	0.001	99.9	26	MRI-ESM2-0	-0.592	0.001	99.9
11	inmcm4	-0.581	0.001	99.9	27	NorESM2-LM	-0.566	0.001	99.9
12	ACCESS-CM2	-0.592	0.001	99.9	28	NorESM2-MM	-0.558	0.002	99.8
13	ACCESS-ESM1-5	-0.565	0.001	99.9	29	SAM0-UNICON	-0.724	0.000	100.0
14	CESM2	-0.577	0.001	99.9					
15	CESM2-WACCM	-0.582	0.001	99.9					

For these reasons, we added some text in the Materials and Methods to describe the statistical tests performed on scatterplots and in particular on fig. 4d:

L399-405: ‘For all of the scatterplots, we define the Pearson’s r correlation coefficients as statistically significant when the p -values derived from the two-tailed Student’s t -distribution exceed a 95% probability threshold. In fig. 4d, to test whether any of the models affects the validity of the statistical test, we perform an additional test, in which we leave one model out at the time, and calculate the correlation coefficient across the remaining models: even so, we find that all correlation coefficients are statistically significant, and that the average of them is -0.58 , the same as when we exclude the two outliers in fig. 4d (GISS-E2-1-G and SAM0-UNICON).’

We decided to leave the two values of the correlation coefficients (for all and excluding the two models) as in the original paper, because while there is no obvious reason to exclude any model, those 2 models clearly behave differently from the vast majority of the ensemble (other 28 models). The approach of presenting multiple regression lines, one with all models and others excluding outliers, is taken in many papers (e.g., Woollings et al. 2012).

As requested, below are the plots of the analysis of surface temperature and precipitation excluding SAM0-UNICON and GISS-E2-1-G (for easier comparison, we also add here the related panels of fig. 2 and fig. 3 from the paper). As you will see, conclusions don’t change with the exclusion of these two models. We added a brief description of this additional test to the Material and Methods at L405-407: *‘In addition, if we exclude the two outliers in fig. 4d from the averages in the map plots of figures 2 and 3 for surface temperature and precipitation, we still find very similar results (not shown).’*

For the plots of precipitation and the easterly wind, in panels a and b I would like to see stippling using the more stringent (and relevant in this case) test that 80% of the models agree on the sign of the change relative to that of the ensemble mean (as was done for SST). Since we are making conclusions about the difference between 2 groups of models, how these models are different from the ensemble mean is the appropriate measure.

The differences between the means of the two large and small AMOC decline of the two groups are already shown to be statistically significant using t-tests in panels (c) in the figures in the manuscript (i.e., we test whether the means of the two groups belong to different distributions or not). These tests were supplemented with randomization tests described in the Materials and Methods at L380-385.

On the other hand, the stippling for panels (a) and (b) of the figures, which are the ones referenced here, are not meant to provide statistical significance but a further measure of inter-model ‘reliability’, i.e., whether or not models in the same group show inter-model agreement in the sign of the change compared to the preindustrial climate. While in the previous version of the manuscript, this was defined as agreement on the sign change from the pre-industrial climate, in the revision we decided to improve this definition by defining inter-model reliability when the majority of models agree on the sign of the change with the ensemble mean of the large/small AMOC groups. This helps rule out whether it is outliers driving the sign of the group average, or whether the majority of group members agree on the sign. We continue to use a different test for surface temperature because in response to such large forcing (4xCO₂) all models show a positive temperature change: thus, the plots would just show reliability everywhere, which we deemed not informative. Hence, for surface temperature and AMOC stream-functions in fig. 1 we decided to further refine the definition of inter-model reliability based on the difference of the change from the median value change of all models, as in the previous version of the manuscript.

We revised the manuscript at L386-398 to improve clarity on these definitions: *‘We further provide measures of inter-model ‘reliability’ to assess whether within the large and small AMOC decline groups there is good inter-model agreement in the changes from the pre-industrial control climate. We use two definitions. In panels (a) and (b) of fig. 3 and fig. 4, reliability is indicated by the ‘/’ stippling and is defined where at least 80% of the models in each group agree in sign with the ensemble mean of the group for each grid point. This definition is informative when the expected change could be either positive or negative. However, for other variables, such as surface temperature, the change in response to 4xCO₂ forcing is almost always of the same sign for all the models. For this reason, in panels (a) and (b) of fig. 1, 2, S3 and S4, we use a more stringent definition of reliability, indicated with the ‘x’ stippling. In this case, the x-stippled areas show where at least a certain percentage of models fall on either side of the median of all the CMIP5+CMIP6 models (i.e., whether the majority of models is above or below the median change). The specific threshold for the ‘x’ stippling depends on the variable: for fig. 2, S3 and S4 is 60%, while for fig. 1 is 80%.’*

I have also included some comments on the attached pdf. No need to respond to these comments separately as the important points are detailed in my written review.

For completeness, we report here the responses to the major comments that were raised in the marked up PDF, but absent from the written review. We also addressed all the other smaller comments in the PDF, but we did not report those below.

L38: Changed ‘largest scale oceanic currents’ to ‘large system of ocean currents’

L43: Changed ‘largely contributes’ to ‘is thought to’

Comment at L72: Does this study really address why AMOC projections are uncertain or provide any means of narrowing uncertainties?

We do not claim this in our study, hence we removed the following statement: *‘In order to narrow down the inter-model range in projections of future climate change, it is crucial to identify the sources of uncertainty and their impacts. Hence, it is of primary interest to quantify the inter-model spread due to the AMOC response.’*

We modified the text at L71-77 to provide a more accurate motivation for our study: *‘Even though an AMOC shutdown within the next century is deemed unlikely^{27,42}, there is a wide range in the projected AMOC decline rates^{43,44,45,28}, and the consequences of the inter-model spread in the AMOC response, including those on the NAWH, are uncertain. In this study, we investigate whether*

changes in global surface temperatures, precipitation and large-scale wind circulations, are related to the inter-model spread in AMOC decline rates by examining an ensemble of 30 abrupt-4xCO2 simulations from the Coupled Model Intercomparison Project phase 5 ('CMIP5')⁴⁶ and phase 6 ('CMIP6')⁴⁷.

Comment at L276-278: Yes, this is a major caveat, but it is not considered carefully in the paper. The origin of the changes in SST, precipitation, and the winds are described as causally related to AMOC changes.

We have provided a longer discussion of this caveat at L305-312 in the revised manuscript: *'The main caveat of our study is that it remains difficult to fully separate the influence of the AMOC from other processes using the existing experiments in the CMIP5 and CMIP6 archives. Future work should focus on refining a method to better isolate the AMOC from other feedback mechanisms, which could include ad-hoc simulations and a coordinated model inter-comparison. Having here laid the foundation that the AMOC response is related to specific impacts in projections of future climate change, additional work is needed to mechanistically explain the links between the AMOC, the surface temperature change and the associated effects on precipitation and atmospheric circulations.'*

In addition, throughout the paper, we have in several instances discussed how the changes in these variables are linked to the AMOC response, including comparisons with previous work in which ad-hoc simulations explored exclusively the role of the AMOC (many references are cited), and also with statistical testing (randomization and t-tests that are described in the Materials & Methods). These tests demonstrate that the link to the AMOC response isn't casual.

L283: Removed 'small'

Reviewer #2 (Remarks to the Author):

Review on "Future climate change scenarios shaped by inter-model differences in Atlantic Meridional Overturning Circulation response" by Bellomo et al.

The authors study global climate impacts of varying degrees of AMOC slowdown under external climate forcing. The methodology of contrasting strong vs. weak AMOC decline in CMIP5 and CMIP6 models is somewhat novel, even though the conclusions are not too surprising. I think this study would advance our understanding on how AMOC might regulate global climate response to external climate forcing. I would recommend its acceptance for publication once the specific comments (see below) are addressed.

Thank you for your helpful feedback. Please see our replies to your specific comments below. We also modified the manuscript, in particular the 'Introduction' and 'Discussion' sections, to better motivate our research question, and provide a better discussion of the novelty and implications of this work.

Specific Comments

Line 129-131, "Fig. 2c confirms that the change in surface temperature is drastically different between the two groups, especially in the North Atlantic and the Arctic, but also in the Southern Ocean in the areas of deep-water formation. ":

Besides what you have described, two other features on Fig. 2c seem interesting:

1) The contrast of warming and cooling anomalies, respectively, over the North America Continent (NAC) and the subpolar North Atlantic Ocean (SPNA). What mechanism do you think may have caused this pattern? All else being equal, if the NAC warms up and cold air outbreaks from the continent weakens, AMOC would decline in response to the resulting weakened surface buoyancy loss in the SPNA, which then leads to the warming hole in the SPNA. This potentially could explain the land-sea contrast seen in Fig. 2c, but in this case, the ultimate driver of the pattern in Fig. 2c is not AMOC but surface buoyancy anomaly in the SPNA. Do you think this might be what's going on here?

For what concerns the mechanism involving the warming/cooling contrast over NAC and SPNA, we think that this generally happens as a result of enhanced warming over land (e.g., Sutton et al. 2007) and we can see it worldwide both in panel (a) and (b) of fig. 2. The reason why the contrast is bigger in the larger AMOC decline models (fig. 2c) could be related to a larger reduction of cold air outbreaks as you suggest, which would ultimately lead to the emergence of the NAWH in the models where the AMOC decline is larger.

In the previous version of the manuscript, we cited articles that provided strong evidence for the AMOC to be the cause of the NAWH. More recent studies have investigated also the role of other forcings, including atmospheric processes. The agreement still seems to be that the AMOC plays a larger role than atmospheric surface fluxes in driving the NAWH (e.g., Keil et al. 2020, Menary and Wood 2018, and references therein), especially in simulations of future climate change. This seems to be supported by the fact that in slab-ocean models, in which ocean heat transport is prescribed to not change from year to year, the NAWH doesn't emerge in the ensemble mean of 2xCO₂ simulations in the CMIP3 archive (Woollings et al. 2012) and it's much smaller compared to a fully-coupled run in a 1pctCO₂ simulation using the ECHAM6 atmospheric model (Keil et al. 2020).

However, since this seems to be a topic under debate, in the revised version of the manuscript we changed the text in the 'Introduction' to provide a better overview of the possible mechanisms leading to the emergence of the NAWH, this time accounting also for atmospheric processes. See revised text at L63-70: *'A recent study³⁶ has investigated the role of atmospheric processes that could alone lead to the formation of the NAWH. Land-sea warming contrast could also explain in part the temperature anomaly difference³⁷. Nevertheless, it seems that in climate model projections of future climate change, ocean circulation plays the largest the role leading to the onset and development of the NAWH^{17,36,38,39}. The NAWH, by changing the baroclinicity of the atmosphere, could affect the large-scale atmospheric response to global warming^{37,40,41}; however, the precise impacts are unknown since there is large inter-model spread in the projections of the NAWH anomaly and its spatial extent³⁸.*

2) The cold anomalies in the Weddell Sea and Ross Sea in the Southern Ocean under stronger AMOC declines. AMOC decline is often associated with bi-polar adjustment in the SST/surface air temperature, that is, warming of the SO and cooling of the NH polar region when AMOC weakens. This picture would not fit with what's shown in Fig. 2c, where the zonal mean surface air temperature anomaly in the SO is negative when AMOC weakens more. What's the reason(s) behind the cold anomalies in the SO?

We believe that these differences are due to the fact that in our study the AMOC decline is driven by the 4xCO₂ forcing. In fact, as you see from the figure below, the delayed warming in the deep-water formation regions seems to be related to a decrease of the mixed layer depth. The figure below shows the difference of the mixed layer depth change between the large and small AMOC decline groups (counterpart of panels (c) in the paper except there are only 8 out of the 10 large

AMOC decline models that made available the mixed layer depth variable, and 7 out of the 10 models for the small AMOC decline group).

We added L134-138 to discuss this point in the revised manuscript: ‘*In the Weddell and Ross seas in the Southern Ocean we see a delayed warming when the AMOC decline is larger (fig. 2c), which, similarly to the North Atlantic, is related to a decrease of the mixed layer depth (not shown). This suggests that in models in which the AMOC decline is larger, there is a more pronounced convection decrease also in the other deep-water formation regions associated with the global thermohaline circulation.*’

We note that we do retrieve the see-saw North Atlantic/South Atlantic pattern as shown in the following figures from Jackson et al. 2015 (left) and Zhang and Delworth 2005 (right) (to be compared with our fig. 2c). Note though, that in those papers there is no increase in CO₂ and the only thing that is changing is an imposed AMOC shutdown in the North Atlantic, while this is not the case in our study. Note, however, that even in their simulations the signal in the Southern Ocean is small.

Minor comments

Line 359-361, “which is defined where at least a certain percentage of models in each group agree on the sign of the difference between each model’s change and the median change of all CMIP5+CMIP6 models.”: At first, I was confused by this statement. It might be helpful to state that this method identifies if the majority of models in the large or small AMOC decline group fall on one side of the median of all the models.

We modified the related paragraph to improve the clarity of the text. See revised manuscript at L394-398.

Clarification

Line 643. “ The ‘*’ stippling indicates...”; Line 644-646, “Line In panels (a), (b) and (c), superimposed contours show the climatological mean surface temperature computed from all models.” :Fig. 2c stippling is not * but squares. Contours in Fig. 2c are not climatological mean surface temperature - it’s the same as the color shaded field.

Thanks for bringing these errors to our attention: we made the necessary corrections.

Reviewer #3 (Remarks to the Author):

This is one of the most interesting CMIP6 analyses I've seen and I strongly recommend publication. The authors have found that changes in the Atlantic meridional overturning circulation determine a wide range of climate responses associated with global warming. There are strong connections with patterns of surface temperature change, precipitation patterns and zonal winds. This has wide implications on a variety of fields: oceanographers who need to characterize the observed AMOC and its changes; modelers who need to reproduce the observed circulation with more fidelity; and climate impact scientists, who should recognize that precipitation across the globe is affected by the AMOC.

We thank you for your very positive feedback and for highlighting the implications of these results on other fields.

Many of the questions I had while reading were answered via the supplementary information or methods. I have just a few comments. First, it might be worth comparing results in the Southern Hemisphere with Shih-Yu Lee et al 2011 to see if the larger shift in the SH might be associated with that mechanism.

We found the analysis of Lee et al. 2011 really interesting. We added L263-269 to the revised manuscript, in which we propose the mechanism in Lee et al. 2011 as a possible explanation of what we see in the southern hemisphere: *‘ In the southern hemisphere, changes are similar during austral winter (JJA) (not shown). Both in the annual mean and in JJA, we see an intensification of the southern hemisphere westerlies that is more pronounced in the large AMOC decline group (fig. 4c). A mechanism proposed in a previous study⁶³ could explain this difference: the more southward dislocation of the Hadley cell in the large AMOC decline group (fig. 3c) could contribute to further strengthen the southern hemisphere westerlies through the weakening of the subtropical jet. However, a more detailed analysis of the interaction of the eddies with the mean flow is needed to corroborate this idea. ’*

However, we believe that a much more in-depth analysis is needed to prove this mechanism. In fact, while there is no SST change in the southern hemisphere in Lee et al. 2011, we do have some relatively colder anomalies in the Southern Ocean in the large AMOC decline group here (related to a decrease of mixed-layer depth, see L134-138 and response to reviewer #1). Therefore, since a direct comparison is not possible, we think this analysis belongs to a separate study.

Second, I would probably recommend calling the winds westerlies instead of easterlies but there's no loss of generality there.

We now refer to the easterly/westerly wind as ‘zonal wind speed’, believing this is the most appropriate terminology. As suggested, we refer to the winds as ‘westerlies’ when we talk about the mid-latitudes.

Finally, I found the detailed pattern of precipitation quite fascinating, and if there's room in the text I might recommend a little more detailed description of the locations with north/south shifts vs other mechanisms.

We added at L196-198 a short discussion of the thermodynamic vs dynamic mechanisms that could be associated with the precipitation differences, and added a few references. We also added more discussion at L300-304 and L309-312:

L196-198: ‘Overall, in the large AMOC decline group, precipitation changes driven by the change in the SST pattern and dynamic changes in atmospheric circulation seem to dominate over the thermodynamic changes related to the increase in specific humidity due to global warming^{54,55,56}.’

L300-304: ‘In addition, our results also suggest that more caution should be used when investigating mechanisms of climate change related to precipitation and large-scale atmospheric circulation patterns. In fact, it seems that some of the uncertainties in precipitation and wind speed, which were previously associated to dynamic changes in wind circulation^{54,55}, may be ultimately driven by the inter-model spread in AMOC response and its influence on the NAWH³⁹.’

L309-312: ‘Having here laid the foundation that the AMOC response is related to specific impacts in projections of future climate change, additional work is needed to mechanistically explain the links between the AMOC, the surface temperature change and the associated effects on precipitation and atmospheric circulations.’

REVIEWER COMMENTS

Reviewer #1 (Remarks to the Author):

This paper is much improved. I appreciate the through responses to my comments as well.

I still find that the paper makes a number on statements on causality that are too strong. The abstract is actually spot on, making careful statements regarding associations between AMOC changes and changes in other climate variables. This care is needed because an association between variables cannot show that a difference in a surface climate variable is "driven" by the AMOC. Throughout the rest of the paper, the authors are not as careful, stating that the AMOC drives various other changes (e.g., line 165, 291, 301-303). These statements should be revised, taking the same care for accuracy as was made on the abstract. The caveat that it is difficult to untangle mechanisms is mentioned at the very end of the paper (line 304-305), but this need to be made clear earlier.

Surface temperature changes:

To say that the pattern of the difference between large minus small AMOC decline models and those seen in water hosing experiments "strikingly similar" seems like a bit of an overstatement. The SST patterns in both Zhang and Delworth (2005) and Jackson et al (2015), two of the papers cited here, are not restricted to the subpolar gyre and Arctic---large negative SST anomalies occur in the NH tropics. Furthermore, in both these models there are positive SST anomalies in the southern hemisphere, likely related to increased ocean heat transport from the SH to the NH. Figure S3 does not show this feature.

What causes the large warming over Canada (just to the west of the North Atlantic warming hole) that is seen in the difference in normalized surface temperature change between the large and small AMOC decline models? This feature seems to depend on the normalization by the global mean temperature change in each model since it is absent when the temperature changes are not normalized by the global mean temperature change (panel c of Figure S3).

Precipitation changes:

One of the main "global impacts" related to the AMOC is supposed to be shifts in the ITCZ, yet in the zonally averaged plot, the zonal mean shifts are only shown for the Atlantic basin. Can you show the zonal mean shifts over all ocean basins as well? Are they similar to what is seen in the Atlantic? Also, what is the cause of the changes in the extratropical precipitation that is shown in Figure 3d?

Wind changes

I found the wind changes to be the least compelling section of the paper. The relationship between the AMOC changes and the change in the latitude of the jet is not very strong ($R^2=0.33$). More importantly, the physical mechanism seems less robust than the mechanisms proposed for SST and precipitation. The argument is the strong AMOC decline models have less Arctic amplification and thus a greater role for tropical heating and a jet that is shifted polewards. However, I would argue that there are means other than the AMOC that impact Arctic amplification in a model under CO₂ forcing, such as the amount of ice in the original mean state. While this paper focuses almost solely on differences in variability, there are differences in means are present between different climate models. It is already mentioned that the mean AMOC plays a role in the AMOC decline (strong AMOC models have larger declines).

I have a number of specific comments in the pdf, but they do not require response as all the main points are summarized above.

Martha Buckley

Reviewer #2 (Remarks to the Author):

I thank the authors for addressing the questions I raised in the previous review. I only have a few minor

comments (listed below). I recommend its acceptance for publication.

Line 132-133, "... and the Arctic":

By Arctic you mean the Nordic Sea?

Line 133, "delayed warming":

What you showed is "weakened" warming. No timing delay was studied.

Line 138, "absolute near surface air temperature":

How is this defined? Obviously, it's different from "surface temperature" (SST and 2m air temperature).

Line 167-168, "the AMOC decline acts as a regional negative feedback, but only in the models featuring a large AMOC decline":

The negative feedback kicks in as long as AMOC decreases, but its effect is large (small) in models with large (small) AMOC decrease. In other words, it is amplified in models with large AMOC decline. Please clarify.

Line 282-293, "this effect is absent and there is actually a relatively larger warming in the SPNA." :

See my comment above. I don't think it's "absent" in small AMOC decrease models, even though its effects are weak in these models.

Reviewer #3 (Remarks to the Author):

The authors have done an excellent job responding to all my comments and those of the other reviewers. I recommend publication.

We thank all reviewers and the editor for providing another round of revisions. Below you can find the replies to each of the reviewers' comments. In addition to the replies below:

- we made some small changes to the introduction to improve readability
- we removed the word 'scenarios' from the title because we thought it could be misinterpreted as if we were analyzing 'ScenarioMIP' simulations while we are not (the abrupt-4xCO₂ experiment belongs to 'CMIP')
- we removed the regression line from fig. S2 ('Dependence of global mean air temperature change on AMOC change') because the correlation coefficient is not statistically significant

Reviewer #1 (Remarks to the Author):

This paper is much improved. I appreciate the through responses to my comments as well.

We thank Dr. Martha Buckley for providing another round of revisions and useful feedback.

I still find that the paper makes a number of statements on causality that are too strong. The abstract is actually spot on, making careful statements regarding associations between AMOC changes and changes in other climate variables. This care is needed because an association between variables cannot show that a difference in a surface climate variable is "driven" by the AMOC. Throughout the rest of the paper, the authors are not as careful, stating that the AMOC drives various other changes (e.g., line 165, 291, 301-303). These statements should be revised, taking the same care for accuracy as was made on the abstract. The caveat that it is difficult to untangle mechanisms is mentioned at the very end of the paper (line 304-305), but this need to be made clear earlier.

To address these concerns we made several changes throughout the text. The major ones are reported below:

Old text at L165-166:

'In fact, the key finding here is that the inter-model spread in the AMOC response is a major driver of uncertainty in SPNA SST change.'

→ Modified text at L174-175:

'Given this strong relationship, we argue that the inter-model spread in the AMOC response is a major source of uncertainty in SPNA SST change.'

Old text at L290-292:

'The differences in the response of large-scale atmospheric circulation and precipitation are also partly explained by the reduction in Arctic amplification caused by a larger AMOC decline.'

→ Modified text at L310-313:

'The differences in the response of large-scale atmospheric circulation and precipitation are partly explained by differences in Arctic amplification. Even though Arctic amplification also depends on other factors²⁷, we find that models in which the AMOC decline is larger exhibit a reduced Arctic amplification.'

Old text at L301-303:

'In fact, it seems that some of the uncertainties in precipitation and wind speed, which were previously associated to dynamic changes in wind circulation^{54,55}, may be ultimately driven by the inter-model spread in AMOC response and its influence on the NAWH⁹.'

→ Modified text at L322-324:

'In fact, it seems that some of the uncertainties in precipitation and the jet stream, which were previously attributed to dynamic changes in wind circulation^{54,55}, are also influenced by the inter-model spread in the AMOC response and its impact on the NAWH³⁹.'

Old text at L312-315:

'In conclusion, the key finding of this work is that the inter-model spread in the AMOC response drives different climate change scenarios in a number of societally important atmospheric variables. The implication is that not only is ocean circulation important for climate change, but also there is large uncertainty arising solely from the inter-model spread in the AMOC response.'

→ **Modified text at L333-337:**

'In conclusion, the key finding of this work is that the inter-model spread in the AMOC response is linked to the uncertainty in the projections of a number of societally important atmospheric variables. The implication is that not only is ocean circulation important for climate change, but also that the uncertainty in the AMOC response may amplify the inter-model spread in the projections of temperature, precipitation and large-scale wind circulation.'

To address the criticism that the caveat is only discussed at the end of the paper in the Discussion section, we added new text at **L118-122**, right before we discuss the results on the changes in the atmospheric variables that we relate to the differences in the AMOC decline: *'Finally, we note that since in these experiments the AMOC isn't artificially modified, but rather changes in response to the increase in CO₂, we are unable to fully separate the impacts of the AMOC from other forcings; hence, we rely on statistical tests to support our findings (see Materials and Methods).'*

Surface temperature changes:

To say that the pattern of the difference between large minus small AMOC decline models and those seen in water hosing experiments "strikingly similar" seems like a bit of an overstatement. The SST patterns in both Zhang and Delworth (2005) and Jackson et al (2015), two of the papers cited here, are not restricted to the subpolar gyre and Arctic---large negative SST anomalies occur in the NH tropics*. Furthermore, in both these models there are positive SST anomalies in the southern hemisphere, likely related to increased ocean heat transport from the SH to the NH. Figure S3 does not show this feature**.

We removed 'strikingly similar' from the text, and to address these concerns we modified the text as follows:

Old text at L138-145:

'In addition, the difference in absolute near surface air temperature change between the large and small AMOC decline groups (fig. S3), which is not divided by Δ GSAT, is strikingly similar to the response to a disruption of the AMOC in water hosing model experiments^{13,14,15}, with the exception of the Weddell and Ross seas as noted above, which, however, are not modified in the water hosing experiments. The similarity between these earlier experiments and our results further supports the hypothesis that the differences in the patterns of surface temperature change shown in fig. 2 are largely influenced by the difference in AMOC decline between the large and the small AMOC groups, rather than by other processes.'

→ **Modified text at L144-154:**

'In addition, the difference in near-surface air temperature change (not divided by Δ GSAT) in the extra-tropical North Atlantic between the large and small AMOC decline groups (fig. S3) is very similar to the response to a disruption of the AMOC in water hosing model experiments^{13,14,15}. Differently from the water hosing experiments, we don't see a relative

warming in the Southern Ocean and in the Weddell and Ross seas, which is due to the fact that in the abrupt-4xCO2 simulations the forcing is globally uniform and very different in nature from the freshwater anomaly added to the North Atlantic basin in the water hosing experiments. Nevertheless, despite the very different experimental designs, the similarity between these earlier experiments and our results further supports the hypothesis that the differences in the patterns of surface temperature change in the North Atlantic shown in fig. 2 are largely influenced by the difference in AMOC decline between the large and the small AMOC groups, rather than by other processes.'

As you can see above we specified in our revised statement that we are referring to the 'extra-tropical North Atlantic' when we argue that changes are 'very similar'. Also at L153 we specify 'North Atlantic' when discussing similarities with the water hosing experiments. We further discuss why we don't expect to see the exact same patterns of change as in the water hosing experiments (the forcings are very different).

To respond to the other two comments: *Note that both in fig. 2c and S3c there are negative anomalies in the NH tropics also in our paper, albeit weaker. **We do find anomalies of opposite sign as in the aforementioned papers in the SH tropics in fig. 2c, albeit weaker.

What causes the large warming over Canada (just to the west of the North Atlantic warming hole) that is seen in the difference in normalized surface temperature change between the large and small AMOC decline models? This feature seems to depend on the normalization by the global mean temperature change in each model since it is absent when the temperature changes are not normalized by the global mean temperature change (panel c of Figure S3).

The difference in warming between the NAWH and Canada is larger in the models with larger AMOC decline, mostly because in those models the NAWH is more pronounced (fig. S3). When we normalize each model by their respective change in GSAT, we are actually dividing the models with larger AMOC decline by a smaller change in GSAT (on average; see fig. S2 and Table S2). Instead, we are dividing the models with a smaller AMOC decline by a larger increase in GSAT. For this reason, the warming difference between the NAWH and Canada of the models with a larger AMOC decline is emphasized in fig. 2c compared to fig. S3c. In fig. 2c, because models are normalized, we are emphasizing the role of the AMOC decline, compared to fig. S3 where Δ GSAT (hence, the associated global feedback) is not accounted for.

The reasons we decided to normalize are provided in the text at L113-115 and in the Materials & Methods at L388-390, and were not changed from the previous version of the manuscript. Related to this, note that we also (already) speculated at L155-158 on the fact that the effect of the AMOC on temperature does not seem to be zonally uniform in the northern hemisphere, as instead suggested in an earlier study. Hence, we do not think that we need to make additional changes to address this point.

Precipitation changes:

One of the main "global impacts" related to the AMOC is supposed to be shifts in the ITCZ, yet in the zonally averaged plot, the zonal mean shifts are only shown for the Atlantic basin. Can you show the zonal mean shifts over all ocean basins as well? Are they similar to what is seen in the Atlantic? Also, what is the cause of the changes in the extratropical precipitation that is shown in Figure 3d?

We made the counterpart of fig. 3d for the global oceans (see below). You can still see a southward shift globally, but this is not statistically significant and also it is dominated by the Atlantic basin.

This can already be seen in fig. 3c. We added the following sentence at L233-235 to discuss the findings of the figure below: *‘If we examine the zonal mean ITCZ change globally (not shown), we still find a southward shift into the southern hemisphere, but this is dominated by the Atlantic contribution and it is not statistically significant at the global level.’*

While it is true that the one of the global impacts of an AMOC shutdown is a southward shift in the zonal mean ITCZ, here we are analyzing experiments in which there is a large CO2 forcing and actually the AMOC per se isn't artificially modified as in the water hosing experiments; hence we do not expect the AMOC to dominate the global response of the zonal mean ITCZ as in the water hosing experiments. In the paper we primarily discuss regions of the world where we can see a statistically significant AMOC influence, this is why we show fig. 3d for the Atlantic only and focus the discussion around the zonal mean precipitation change in the Atlantic sector. To further emphasize that we are focusing on changes that are significant, especially over the Atlantic Ocean we modified the following text in the Discussion:

L285-287 old text: *‘In the models with larger AMOC decline, the precipitation over the oceans does not follow the wet-get-wetter/dry-get-drier paradigm, but there is a southward shift of the ITCZ and a reduction in precipitation over the Gulf Stream and the SPNA.’*

→ Modified text at L304-307:

‘In the models with larger AMOC decline, the precipitation over the oceans does not follow the wet-get-wetter/dry-get-drier paradigm: in fact, in the Atlantic there is a southward shift of the zonal mean ITCZ and a reduction in precipitation over the Gulf Stream and the SPNA.’

To answer the second question, we added the following text at L230-233: *‘This, as noted above, could be related to a difference in the relative influence of thermodynamic and dynamic drivers of precipitation change, which are associated with the difference in warming in the North Atlantic between the two groups.’* Note that addressing the mechanisms of precipitation change is not trivial and it is likely that they are very different in the tropics and in the extra-tropics. Note that in the discussion we also (already) had another comment on this (L329-332): *‘Having here laid the foundation that the AMOC response is related to specific impacts in projections of future climate change, additional work is needed to mechanistically explain the links between the AMOC, the surface temperature change and the associated effects on precipitation and atmospheric circulations.’*

Wind changes

I found the wind changes to be the least compelling section of the paper. The relationship between the AMOC changes and the change in the latitude of the jet is not very strong ($R^2=0.33$). More importantly, the physical mechanism seems less robust than the mechanisms proposed for SST and

precipitation. The argument is the strong AMOC decline models have less Arctic amplification and thus a greater role for tropical heating and a jet that is shifted polewards. However, I would argue that there are means other than the AMOC that impact Arctic amplification in a model under CO2 forcing, such as the amount of ice in the original mean state. While this paper focuses almost solely on differences in variability, there are differences in means are present between different climate models. It is already mentioned that the mean AMOC plays a role in the AMOC decline (strong AMOC models have larger declines).

Statistical significance:

We did demonstrate that a correlation coefficient of 0.57 (corresponding to $R^2=0.33$) is statistically significant in the first round of revisions. An additional test was also added to the Materials and Methods to show that significance did not depend on any outliers. Plus, we also showed that removing two outliers - which exhibit a very different AMOC response compared to the rest - leads to a correlation coefficient of 0.70 ($R^2=0.48$). Given the fact that dynamical changes in the northern hemisphere mid-latitude atmospheric circulation are highly uncertain in future climate change (for example, compared to better understood thermodynamic changes), a correlation coefficient of 0.70 (or even 0.57) is not irrelevant. In the paper we do not claim that the AMOC can explain all of the inter-model differences, rather we say that reducing the inter-model spread in the AMOC response may help reduce the uncertainty in the projections of other variables. By showing the R^2 , inter-model scatterplots and maps we effectively quantify the uncertainties in these variables that are associated with the AMOC response.

Physical mechanisms:

For what concerns the comment on the mechanisms, we note that the proposed mechanism would explain the changes in the zonal mean thermally-driven jet at 250hPa (and not the eddy-driven jet at lower levels; see L277-280). The thermally-driven upper level jet is expected to be mainly influenced by upper tropical troposphere heating and Arctic amplification (as discussed at L260-266 and references therein). However, since one of the concerns seems to be that the proposed mechanism is not as robust as the AMOC/NAWH relation, we added the following sentence at L276-277: *‘Although statistically significant, this relationship is weaker compared to the relationship between the NAWH and the AMOC response (correlation coefficient of 0.80).’*

Arctic Amplification and mean state:

We rely on statistical testing to show that on average models with larger AMOC decline exhibit reduced Arctic Amplification (fig. S4), and in the revised manuscript we added the following text to emphasize that the AMOC response is not implied as the only factor influencing Arctic Amplification:

L264-266: *‘Fig. 4c shows that the effect of the tropical heating seems to be stronger when there is larger AMOC decline because in these models Arctic amplification is on average reduced compared to the models in which there is a smaller AMOC decline (fig. S4).’*

L310-313: *‘The differences in the response of large-scale atmospheric circulation and precipitation are partly explained by differences in Arctic amplification. Even though Arctic amplification also depends on other factors²⁷, we find that models in which the AMOC decline is larger exhibit a reduced Arctic amplification.’*

We agree that it is possible that there is a further relation among the mean climate, Arctic amplification and mean AMOC strength. Fig. S1 suggests that this relation is not so simple and would require extensive analysis, which is outside the intended scope of this paper. Note that we

already have some discussion regarding the mean climate (see L337-342 and the reference to ref. 34).

I have a number of specific comments in the pdf, but they do not require response as all the main points are summarized above.

We thank you for also providing an annotated pdf. We addressed those minor comments as well in the revised manuscript.

Reviewer #2 (Remarks to the Author):

I thank the authors for addressing the questions I raised in the previous review. I only have a few minor comments (listed below). I recommend its acceptance for publication.

We thank reviewer 2 for going through our revised manuscript and providing additional helpful comments.

Line 132-133, "... and the Arctic":
By Arctic you mean the Nordic Sea?

Yes, we modified the text as suggested.

Line 133, "delayed warming":
What you showed is "weakened" warming. No timing delay was studied.

We changed 'delayed' to 'reduced'.

Line 138, "absolute near surface air temperature":
How is this defined? Obviously, it's different from "surface temperature" (SST and 2m air temperature).

'Absolute' means that it isn't divided by ΔGSAT (as it was written in the same sentence). Since it seems that this terminology wasn't clear, we removed the term 'absolute' from the revised text, and changed it as follows at L144-146: ***'In addition, the difference in near-surface air temperature change (not divided by ΔGSAT) in the extra-tropical North Atlantic between the large and small AMOC decline groups (fig. S3) is very similar to the response to a disruption of the AMOC in water hosing model experiments^{13,14,15}.'***

We also clarified that the variable plotted in fig. 2 is 'surface temperature' ('ts') and in fig. S3 is 'near-surface temperature' ('tas'). Hence, the following lines were revised:

When describing fig. 2:

L129-130: 'Over the ocean, the surface temperature coincides with the Sea Surface Temperature (SST), while over land it corresponds to the temperature at the surface.'

In the caption of fig. S3:

L865-868: 'Panels (a) and (b) show the annual mean near-surface air temperature (usually at 2 meters) change in the abrupt-4xCO₂ with respect to the pre-industrial control for (a) the average of the large AMOC decline group and (b) the average of the small AMOC decline group.'

Line 167-168, “the AMOC decline acts as a regional negative feedback, but only in the models featuring a large AMOC decline”: The negative feedback kicks in as long as AMOC decreases, but its effect is large (small) in models with large (small) AMOC decrease. In other words, it is amplified in models with large AMOC decline. Please clarify.

We modified this part of the text so that it is line with what is argued here. The revised text is now at L176-178: *In summary, the AMOC decline acts as a regional negative feedback on temperature warming; however, this effect is much larger in the models featuring a large AMOC decline (compare fig. 2a with 2b).*

Line 282-293, “this effect is absent and there is actually a relatively larger warming in the SPNA.” : See my comment above. I don’t think it’s “absent” in small AMOC decrease models, even though its effects are weak in these models.

We modified this part of the text. The revised text is now at L299-302: *The reduced northward heat transport by the AMOC acts as a negative feedback on SST warming in the North Atlantic, which results in a minimum warming in the North Atlantic in the models with stronger AMOC declines. However, in models with weaker AMOC decline, this effect is small and, on average, there is actually a relatively larger warming in the SPNA.*

Reviewer #3 (Remarks to the Author):

The authors have done an excellent job responding to all my comments and those of the other reviewers. I recommend publication.

We thank again reviewer 3 again for their insightful feedback on the manuscript.

REVIEWERS' COMMENTS

Reviewer #1 (Remarks to the Author):

The authors did a great job with the revision. I am satisfied with the response to my comments, and I recommend publication.